# Somatic Alteration Burden Involving Non-Cancer Genes Predicts Prognosis in Early-Stage Non-Small Cell Lung Cancer

**DOI:** 10.3390/cancers11071009

**Published:** 2019-07-19

**Authors:** Dennis Wang, Nhu-An Pham, Timothy M. Freeman, Vibha Raghavan, Roya Navab, Jonathan Chang, Chang-Qi Zhu, Dalam Ly, Jiefei Tong, Bradly G. Wouters, Melania Pintilie, Michael F. Moran, Geoffrey Liu, Frances A. Shepherd, Ming-Sound Tsao

**Affiliations:** 1Princess Margaret Cancer Centre, University Health Network, Toronto, ON M5G 2M9, Canada; 2NIHR Sheffield Biomedical Research Centre, University of Sheffield, Sheffield S10 2HQ, UK; 3Toronto General Research Institute, University Health Network, Toronto, ON M5G 2C4, Canada; 4Program in Molecular Structure and Function, Hospital for Sick Children, Toronto, ON M5G 0A4, Canada; 5Department of Medical Biophysics, University of Toronto, Toronto, ON M5G 1L7, Canada; 6Department of Medicine, University of Toronto, Toronto, ON M5G 2C4, Canada; 7Department of Laboratory Medicine and Pathobiology, University of Toronto, Toronto, ON M5S 1A1, Canada

**Keywords:** cancer genomics, prognosis, mutation burden, copy number, patient stratification, cancer immunology, oncogenic pathways, patient-derived xenograft

## Abstract

The burden of somatic mutations and neoantigens has been associated with improved survival in cancer treated with immunotherapies, especially non-small cell lung cancer (NSCLC). However, there is uncertainty about their effect on outcome in early-stage untreated cases. We posited that the burden of mutations in a specific set of genes may also contribute to the prognosis of early NSCLC patients. From a small cohort of 36 NSCLC cases, we were able to identify somatic mutations and copy number alterations in 865 genes that contributed to patient overall survival. Simply, the number of altered genes (NAG) among these 865 genes was associated with longer disease-free survival (hazard ratio (HR) = 0.153, *p* = 1.48 × 10^−4^). The gene expression signature distinguishing patients with high/low NAG was also prognostic in three independent datasets. Patients with a high NAG could be further stratified based on the presence of immunogenic mutations, revealing a further subgroup of stage I NSCLC with even better prognosis (85% with >5 years survival), and associated with cytotoxic T-cell expression. Importantly, 95% of the highly-altered genes lacked direct relation to cancer, but were implicated in pathways regulating cell proliferation, motility and immune response.

## 1. Introduction

There have been extensive studies evaluating prognostic markers for early stage non-small cell lung cancer (NSCLC) patients. These studies included genes/proteins that are overexpressed in tumor compared to non-tumor lung tissue, mutations in specific “driver” oncogenes and tumor suppressor genes, and gene expression profiles [1,2,3]. To date, no single gene/protein has been shown consistently to be prognostic especially in large multi-institutional patient cohorts [4,5,6,7]. While many prognostic mRNA signatures have been identified by various investigators, few of these signatures have provided meaningful insight into the biological mechanisms that drive their prognostic significance [8,9]. More recently, whole genome and targeted sequencing studies have observed a relationship between the mutational load of tumors and prognosis in NSCLC [10,11,12,13,14]. However, these studies have yet to show consistent association between mutation burden and survival across multiple cohorts. Since some non-synonymous mutations generated neoantigens that induced an anti-tumor immune response [15,16,17,18], they have been reported to predict improved progression-free survival in NSCLC patients treated with immunotherapies [19,20,21]. However, these effects of mutation burden and neoantigens are not well characterized in early-stage NSCLC cases where immunotherapies are currently not given.

We reported previously the genomic, transcriptomic and proteomic profiles of 36 NSCLC patients from their primary tumors and established xenografts [22,23,24,25]. Given NSCLC tumors are genomically heterogeneous [26], we postulated that mutations found in both primary and xenograft tumors are more likely to correlate with patient outcomes. We applied an integrative genomics approach to identify prognostic somatic alterations in the 36 NSCLC patients and validated their predictive performance in multiple, much larger and independent patient cohorts (Figure 1). These results provide new insights into genes previously not implicated in carcinogenesis itself, but that have an impact on patient outcome.

## 2. Results

### 2.1. Somatic Alteration Burden in PDX Is Correlated with Patient Survival

A total of 164–765 genes were characterized with recurrent somatic SNVs or copy number aberration (CNA) in 36 NSCLC patients (Figure 2A). To investigate the importance of these alterations, we correlated their burden to the corresponding patient survival. The comparison of disease-free survival (DFS) and overall survival (OS) of patients with these tumors with ≤343 alterations (first quartile) to patients corresponding to patient-derived xenografts (PDXs) with >343 alterations revealed that the group with the higher number of alterations had significantly better OS (HR = 0.366, 95% CI = 0.140–0.952, *p* = 0.040) and DFS (hazard ratio (HR) = 0.294, 95% CI = 0.103–0.840, *p* = 0.022). The comparison of CNAs alone with DFS showed a weaker and statistically non-significant association (HR = 0.546, 95% CI = 0.185–1.61, *p* = 0.272); this also applies to OS (HR = 0.532, 95% CI = 0.197–1.43, *p* = 0.202). Similarly, the number of somatic mutations alone was not significantly associated with DFS (HR = 0.568, 95% CI = 0.179–1.79, *p* = 0.335) or OS (HR = 1.41, 95% CI = 0.407–4.76, *p* = 0.592).

Across the 36 tumors, a total of 3236 unique genes with somatic alterations detected in more than a single patient were used to perform a penalized regression that was most associated with OS. The regression model highlighted 865 genes with a non-zero contribution to the overall risk of death (Appendix A). Remarkably, only 47 of these genes (5.4%) were identified in the Cancer Gene Census [27]. The 36 patients were then divided into two groups based on the number of altered genes (NAG) among the 865 genes. Patients with a high NAG had better survival (HR = 0.153, 95% CI = 0.051–0.459, *p* = 1.48 × 10^−4^) than patients who had a low NAG (Appendix A) and their associated clinical-pathological features (summarized in Appendix A and individually listed in Appendix A).

### 2.2. Validation Using the Gene Expression Signature Associated with Somatic Alteration Burden

We hypothesized that the prognostic differences of NAG may be recapitulated in the differentially-expressed gene signature between the high and low NAG groups, and this may be used as a proxy of NAG for further validation in publicly-available independent patient cohorts accessioned in the Gene Expression Omnibus web-based data repository (GSE42127, GSE50081and GSE68465 described in Appendix A). We identified 79 genes with differentially-expressed mRNA (>2 absolute fold change, *p* < 0.05) between the two groups in our cohort of 36 patients based on the NAG (Appendix A). Using mRNA expression profiles of the 79 genes from 127 NSCLC patients in GSE42127, the expression scores associated with survival were calculated. Of the 79 genes, only 23 genes had non-zero coefficients when we used penalized regression for fitting to OS in this cohort. The resulting NAG expression signature was significantly prognostic for three independent early-stage cohorts with mRNA expression data (GSE42127 HR 0.23, 95% CI = 0.125–0.422, *p* = 2.3 × 10^−7^; GSE50081 HR = 0.562, 95% CI = 0.332–0.952, *p* = 0.029; GSE68465 HR = 0.518, 95% CI = 0.338–0.794, *p* = 2.2 × 10^−3^ (Appendix A)).

### 2.3. Validation in TCGA Datasets

To test the alteration burden together with the expression signature as a prognostic classifier for NSCLC, we attempted to predict the overall survival of 221 lung adenocarcinoma (LUAD) and 173 lung squamous cell carcinoma (LUSC) cases from TCGA. We counted the NAG among the 865 genes and calculated the mRNA expression risk score for each TCGA patient tumor, then classified the patients using the same cut-offs that were used to classify our cohort of 36 patients. All patients with a high NAG (at least 87 altered genes) or a low expression risk score had a relatively better prognosis (Figure 2B; HR = 0.575, 95% CI = 0.382–0.870, *p* = 0.0075); this also held true for stage I patients (Figure 2C; HR = 0.391, 95% CI = 0.222–0.685, *p* = 6.9 × 10^−4^). Multivariate survival analyses of stage I patients adjusted for age, sex, smoking history and histology confirmed that the number of alterations was an independent predictor of good OS (HR = 0.407, 95% CI = 0.211–0.787, *p* = 7.7 × 10^−3^, Table 1).

### 2.4. Immunogenic Mutations Correlate with the Best Survival and Cytotoxic T-Cell Signature

Given that alteration burden is associated with good survival outcome, we further hypothesized that some of the mutations in the 865 genes may induce an immune response towards the tumor through the expression of immunogenic peptides or neoantigens. We examined whether the somatic variants within the 865 genes found in TCGA NSCLC patients could encode for tumor-specific antigens presented by MHC Class I molecules. A subset of 86 patients had at least one expressed mutated peptide with high MHC I binding affinity against both HLA-A alleles, and this immunogenic group had significantly better OS (Appendix A; HR = 0.536, CI = 0.341–0.8419, *p* = 0.0068). Among NSCLC stage I patients, 47 were classified as having immunogenic mutations and had significantly better OS (Figure 3A; HR = 0.266, CI = 0.1068–0.6619, *p* = 0.0044), even after adjusting for other clinical factors (Table 2). In comparison, patients stratified using all possible neoantigens, beyond those in the panel of 865 genes, had no significant difference (*p* < 0.01) in overall survival (Appendix A). By combining this immunogenicity classifier with the alteration burden based on NAG, we were able to identify patients with extremely good prognosis. Stage I patients with both immunogenic and high NAG tumors had significantly better survival than patients with either low or high NAG alone (Figure 3B; HR = 0.0807, CI = 0.02315–0.2813, *p* = 7.8 × 10^−5^).

The RNA expression profiles from these two groups of patients were analysed further for immune components. Immune cell population estimates from TIMER [28] revealed moderately, but not significantly higher proportions of B-cells, dendritic cells, and CD8 T-cells in high NAG patient tumors with immunogenic peptides compared to those without neoantigens (Figure 3C). Focusing specifically on markers of cytotoxic T-cells in Figure 3D, we also found in high NAG tumors with immunogenic peptides significantly higher (*p* < 0.05) expression of CD8A, granzyme B (GZMB) and perforin (PRF1). Previously, cytokines CCL5, CXCL9, CXCL10 and IL16 were found to be associated with cytotoxic T-cells [29]. We found CCL5, CXCL9 and CXCL10 to be significantly more highly expressed in our group with immunogenic peptides. In contrast, there were no consistent differences in immune cell populations or cytotoxic T-cell markers between high and low NAG patients without immunogenic peptides (Appendix A).

### 2.5. Prognostic Alterations Enriched in Genes Involved in Extracellular Signaling

To investigate the alternate or additional mechanisms by which alteration burden might be impeding cancer progression, pathway enrichment analyses were performed on the 865 genes. The most enriched pathways involved BARD1 signaling, Janus kinase activity, integrin signaling and extracellular matrix interactions (Figure 4A; hyper-geometric test *p* < 0.01). Within these pathways, a number of genes were altered more frequently across TCGA cases in the group of low-risk patients with better survival (Figure 4B). Inactivating alterations, which deactivate Wnt signaling, were most frequent in *CTNNB1* (23.1%) and *WNT5A* (22.8%), along with copy gains in the tumor suppressors *NHERF1* (16.4%) and *NF2* (13.1%). The high frequency of copy gains occurring in *ITGA10* (20.9%), *COL20A1* (17.9%) and *COL8A1* (26.5%) suggested that cell adhesion to the extracellular matrix may be affected. These highly-altered genes and pathways have been known to control specific cellular functions such as proliferation, motility and adhesion.

In order to evaluate the effect of these pathway alterations on cell functionality, we selected three NSCLC cell lines (H1573, A549, HCC827) with different levels of NAG and expression risk scores (Figure 5A). The three cell lines had different patterns of expression for the NAG genes we found frequently altered in TCGA cases (Figure 5B). Cell line H1573 had 45 NAG, while A549 and HCC827 had 13 and 18, respectively. HCC827 had the fastest rate of proliferation over 80 h of growth and the highest number of invading cells across a Matrigel membrane after 48 h (Figure 5C,D). In contrast, H1573, which had the highest NAG, had a much slower rate of proliferation and invasion (*p* = 0.0145). Furthermore, HCC827 moved the fastest in a 24-h motility assay compared to the other two cell lines (Figure 5E).

## 3. Discussion

We have shown that the number of somatic alterations among 865 genes is associated with better prognosis in early stage NSCLC patients. These prognostic alterations included not only non-synonymous mutations, but also copy number alterations, and they have a corresponding gene expression signature derived from differentially-expressed genes between high and low NAGs validated in three cohorts of early stage NSCLC. For stage I TCGA patients, we further showed that the integrative NAG score was able to classify patients into three distinct subpopulations with low NAG, high NAG and high NAG with neoantigens. These subpopulations were distributed equally in their frequency of clinical features (Appendix A). Pathway analysis of the panel of 865 genes revealed that the majority of the mutations had not been implicated in driving cancer pathogenesis, but were associated with processes relating to immune activation, tumor cell–extracellular matrix interactions and cell motility, which were supported by cell line experiments. These findings provide evidence that high-level somatic genomic aberrations involving non-cancer-associated genes may confer better prognosis through the generation of neoantigens, impairment of cellular functions involved in tumor cells and stroma interaction and motility.

Few recent investigations of mutation or neoantigen burden in early stage or untreated lung cancers provided contradictory evidence for the directionality of the relationship between burden and survival [10,11,12]. These studies relied on measuring burden by counting all point mutations across genomes and, thus, were sensitive to mutation calling error (Appendix A). In contrast, our burden measure of counting the number of genes with at least one alteration (point mutation, CNA, neoantigen or RNA expression) may be more robust to variant calling errors and variation in the number of variants within each gene.

Smaller targeted gene panels recently have been shown to be clinically effective at measuring overall tumor mutation burden in mostly known cancer driver genes [10,15]. Our observations indicate that mutation burden in non-driver genes may also have an effect on patient outcome. The mechanism by which alteration burden in gene sets affects cancer progression and patient survival remains unclear, but it has been hypothesized that excessive genomic instability can lead to deleterious mutations and gene losses that impede tumor growth [30,31]. Greater DNA damage, especially due to dysregulation of DNA repair mechanisms, could result in deleterious mutations or copy number losses in many of the 865 genes that are associated with cell adhesion, motility and integrin signaling, cellular functions that are likely also essential in the phenotypic manifestation of malignant tumor cells [32,33]. Integrin signaling is a major pathway contributing to cancer cell survival and has been a target for antagonists to inhibit tumor growth [34]. Therefore, the occurrence of copy loss alterations in integrin pathway genes may mitigate tumor cell malignancy and is thus a marker for less aggressive tumors in patients with good survival outcomes.

Somatic mutation burden has been correlated with neoantigen load and T-cell infiltration in NSCLC and melanoma [14,35,36,37], yet there is a lack of evidence for this relationship in early-stage cancers, which are predominantly untreated with therapeutics. Our assessment of neoantigen load among the 865 genes is the first to show that even in stage I NSCLC cancer without immunotherapy, the presence of neoantigens and high mutation burden is correlated with T-cell expression. Previously, mutation burden and neoantigen load were used to identify patients with good prognosis following immunotherapy treatment, such as immune checkpoint inhibitors, but there has been debate about whether both markers are needed given that they select for a similar subgroup of patients [14,15,38]. Even though all the patients we identified with immunogenic mutations (neoantigens) were also classified as high alteration burden and as having long OS (>85% surviving longer than five years), we identified a unique subgroup of patients with low alteration burden and no immunogenic mutations that had far worse OS (median survival of two years). In contrast to previous reports about the clonality of neoantigens that are prognostic [14,16], the immunogenic mutations we used for patient stratification did not show evidence of being subclonal compared to other mutations based on their variant allele fractions (Appendix A). Our criteria for immunogenicity prediction may have been overly stringent, resulting in far fewer neoantigens found in each patient compared to other studies and may have led to our approach missing some clonal or subclonal neoantigens.

## 4. Materials and Methods

### 4.1. Patient and NSCLC PDX for Genomics Profiling

This study involved 36 patients [22,23,24,39] whose NSCLC tumors were surgically resected and successfully engrafted as PDX models in non-obese diabetic (NOD) severe immune deficient (SCID) mice. Short indels and single nucleotide variants (SNV) were identified from whole exome sequencing in the primary and PDX tumors. Copy number alterations were profiled using the HumanOmni 2.5 BeadChip SNP array and processed in accordance with previous reports [22,23,24,39]. The average log likelihood ratio (LRR) for each gene was used to determine copy gains (LRR > 0.5) and copy loss (LRR < −0.5).

### 4.2. TCGA Patients

Published mutation, copy number, RNA-seq and clinical data were downloaded for LUAD [40] and LUSC [41] patients on cBioPortal, using the cgdsr R package [42], and clinical data were downloaded using the FireBrowseR R package [43]. Copy number gains and losses were determined using an LRR cut-off of ±0.5. RNA expression data in the RNA-seq by expectation maximization (RSEM) scale from RNA-seq were transformed by the asinh function for survival analysis.

### 4.3. Stratification by High-Level Mutation Burden

To identify genomic alterations associated with patient survival from somatic profiles, we first defined a gene as “altered” if there was a somatic non-synonymous mutation, copy number gain or copy number loss. Genes altered only in one patient tumor were removed. The resulting alteration profiles across patients were fitted to the overall survival (OS, up to five years) of the corresponding patients. ElasticNet from the R package “glmnet” Version 1.9-8 was used for the regression fit with alpha at 0.1 and the best fitting model selected from the maximum of the deviance ratio. In the ElasticNet model, 865 genes had non-zero coefficients.

The number of somatic alterations among 865 genes (NAG) was used as the risk score for each TCGA patient. The risk scores were dichotomized at the first quartile to assign patients to one of the prognostic arms (high vs. low number of alterations). The proportions of DFS and OS were calculated using the Kaplan–Meier method, and the difference between curves was tested using the log-rank test. The OS is the time between the date of diagnosis to the date of death or last follow-up, and the DFS is the date of diagnosis to the date of death, or relapse or last follow-up. All reported hazard ratios (HR) scores and *p*-values used the Kaplan–Meier method unless reported as multivariate analyses. A Cox proportional hazards model was used to fit survival times to the number of altered genes while adjusting for other clinical factors including age, stage and smoking status. The significance for the Cox proportional hazards model was based on the Wald test. The R code for conducting this validation test on TCGA datasets can be found at https://github.com/TransAnalytics/cancer_prog_prediction.

### 4.4. Stratification by Gene Expression Profiles Corresponding to NAGs

The mRNA expression profiles in the 36 NSCLC tumors were measured as previously described and accessioned in the Gene Expression Omnibus (GEO) data repository (GSE68929) [22,23,24,39]. 79 differentially-expressed genes (absolute fold change >2 and *t*-test *p*-value <0.05) between the high and low number of genomic alterations were identified. The expression profiles of these genes from the UT Lung Spore dataset (GSE42127) [8] were fitted to the OS using ElasticNet. ElasticNet’s lambda parameter was selected from 5-fold cross-validated likelihood, and the alpha parameter was set at 0.1. The coefficients from the ElasticNet model, which formed the 23-gene prognostic classifier, were summed to create an expression risk score for each patient. An expression risk score ≥0.17 was associated with low NAGs, whereas an expression risk score <0.17 was associated with high NAGs. The cut-off at 0.17 was at the third quartile of risk scores, which was consistent with the first quartile cut-off for NAG. All reported survival differences were tested using the Wald test within the Cox proportional hazards model, and follow-up of patients was measured up to 5 years. Validation of NAG expression risk scores was performed on the Director’s Challenge Consortium for Molecular Classification of Lung Adenocarcinoma dataset with a total of 442 samples (GSE68465) [1] and the UHN181 cohort of 181 NSCLC patients (GSE50081) [44].

### 4.5. Estimating Immunogenicity

Non-synonymous mutations causing amino acid changes were identified among the 865-gene panel for TCGA patients. Protein sequences were acquired from the UniProt reviewed canonical human proteome UP000005640 FASTA file. Windows of 8–11 amino acids in length were derived by applying non-synonymous SNVs to their respective protein sequence and using all 8–11mer amino acid peptides containing the altered amino acid. The 8–11mer altered peptides for each patient were input into NetMHCpan v3.0 [45] along with both of the patient’s supertyped HLA-A alleles to calculate their predicted HLA-A binding affinity values for each altered peptide. The 4-digit HLA types for the patients were sourced from The Cancer Immunome Atlas [46], which used Optitype [47] to call the HLA-A alleles from RNA-seq FASTQ files. The HLA types were supertyped into standard categories prior to carrying out the HLA-A binding affinity prediction analysis [48].

We defined peptides as antigenic if they had an HLA-A binding affinity below 500 nM with both HLA-A alleles. In addition, antigenic peptides’ genes had to be expressed above the median expression level for that gene calculated across all patients in order for the peptides to be considered immunogenic, and the patient HLA-A gene needed to be expressed above the median HLA-A expression level across all patients. Patients were considered to have immunogenic peptides that could potentially trigger an immune response by fulfilling these three conditions: strong binding affinity, high expression of the antigenic peptide to ensure they are clonal and high expression of the MHC-I receptor. Patients were stratified based on whether or not they had above the median number of immunogenic peptides (1 or more) as outlined above, either alone, or in combination with the NAG score. A Cox proportional hazards model was used to calculate the effect size/risk factor associated with the presence/absence of these features, along with the *p*-values for the differences between patient groups.

### 4.6. Pathway Enrichment

Gene set enrichment was performed on the 865 genes using the database for annotation, visualization and integrated discovery (DAVID) and WebGestalt [49]. Using the DAVID API, 120 “adhesion”-, “integrin”- and “extracellular matrix”-related genes were isolated from the initial gene set. Of these, the list was refined further by filtering for those with alteration frequencies greater than 10% among TCGA patients. Pathway memberships of genes were extracted from GeneCards, KEGG and Pathway Commons [50] and visualized by Cytoscape (v3.2.0). Nodes on the edges of the network that did not connect to any of the main cancer pathways were removed (i.e., CLTC and EPN1). The genes were arranged by cellular location into either the extracellular matrix (ECM) and plasma membrane or the cytoplasm.

### 4.7. In Vitro Functional Assays

Cell proliferation assays were performed on A549, H1573 and HCC827 cells as described previously [51]. In brief, 5000 cells were seeded per well of an E-plate. Impedance was measured every 15 min for 80 h. Growth curves were constructed using the xCELLigence platform (ACEA Biosciences, San Diego, CA, USA). Tumor cell invasion was assessed in vitro by the reconstituted basement membrane (Matrigel) invasion assay [52], which was performed using 8-μm polycarbonate filters coated with reconstituted basement membrane (Matrigel; BD Bio-sciences, San Jose, CA, USA). The motility of cells was measured after seeding in a plastic-bottomed 24-well dish and incubated for 12 h. FBS was added to the medium, and the migration assay was carried out for 24 h. Images were acquired every 20 min for approximately 24 h using a 10× phase objective.

## 5. Conclusions

Extensive efforts by many laboratories have attempted to identify cancer genomic features that correlate with the prognosis of NSCLC patients [3,53]. Very few studies have attempted to conduct biological validation of the prognostic genes identified in these prognostic signatures [9,54]. In contrast, Tang et al. used a systems biology approach integrating genome-wide functional (RNAi) data and genetic aberration data to derive a 12-gene prognostic signature that is also predictive for adjuvant chemotherapy benefits in NSCLC. Our study represents a translational research approach to identify genomic features of NSCLC tumor cells that could confer the prognostic implication in patient clinical outcome. We have provided evidence that the good prognostic impact of a high level of genomic aberrations was contributed by genes that are not considered oncogenic drivers in NSCLC through potential neoantigen generation and impairment of proteins that remain essential for tumor cells to manifest their malignant phenotypes.

## Figures and Tables

**Figure 1 cancers-11-01009-f001:**
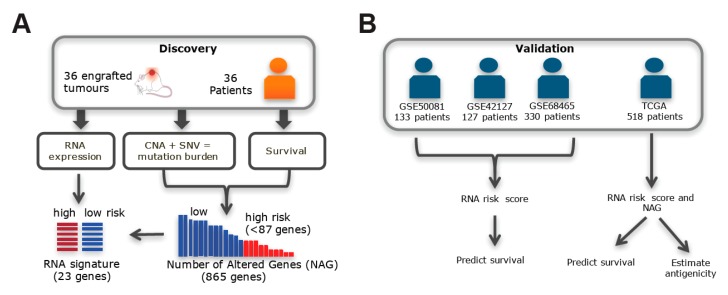
Translation of genomic information from patient-derived xenografts (PDXs) to predict patient outcome. (**A**) Somatic copy number aberrations (CNA), single nucleotide variants (SNV) and gene expression (RNA) were profiled in PDXs. The number of alterations within 865 genes (NAG) and the associated 23 genes’ expression signature were selected using the survival of patients corresponding to the PDXs. (**B**) The expression signature was used to calculate a risk score and predict overall survival (OS) in three independent cohorts of NSCLC patients with data accessioned in the Gene Expression Omnibus web-based data repository (GSE50081, GSE42127 and GSE68465). Survival prediction was also tested for 518 NSCLC stage I patients from The Cancer Genome Atlas (TCGA), by using both the expression signature and NAG. The mutations found in the 865 gene panel of the TCGA samples were also assessed for antigenicity.

**Figure 2 cancers-11-01009-f002:**
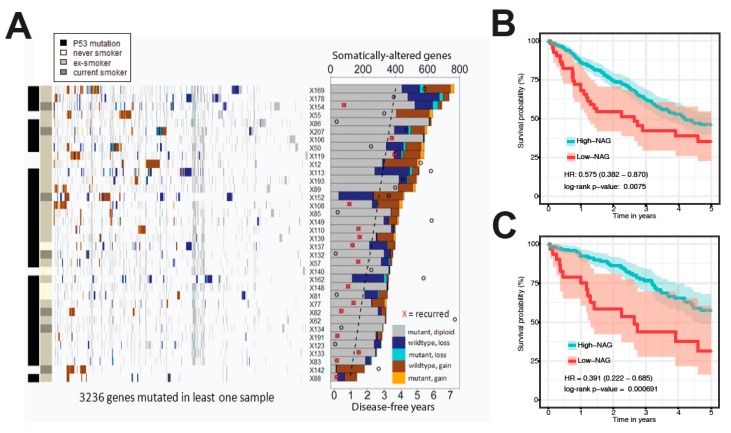
Association between somatic alterations found in PDXs and OS of patients. (**A**) A heatmap of SNVs and CNAs across PDX models in genes that have been altered at least once. The bar graph describes the total number of altered genes in each PDX (labeled by their ID preceded by “X”). Different combinations of SNV point mutations and CNAs are described by the side bar graph. Circles on the bar graph show the disease-free years to last follow-up, and the red “X” denotes the recurrence of disease for each PDX’s matched patient, while the dashed line shows the best fit running through them. Genes were ordered by hierarchical clustering across the samples. Kaplan–Meier curves show significant differences, as evidenced by the hazard ratio (HR), its confidence interval and log-rank test *p*-value, in the overall survival (OS) between high and low NAG in (**B**) all TCGA NSCLC patients and (**C**) only in stage I patients.

**Figure 3 cancers-11-01009-f003:**
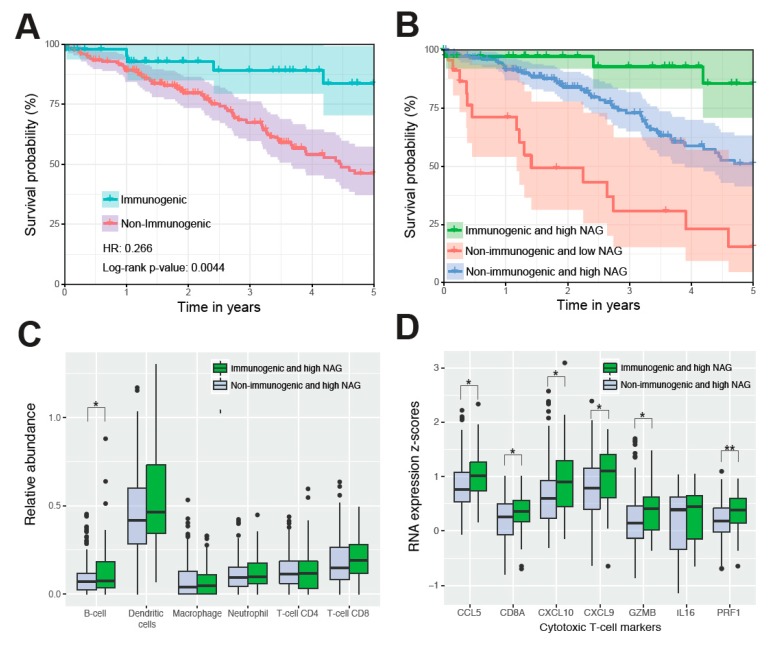
TCGA NSCLC patients stratified by immunogenic neoantigens. (**A**) Stage I patients (47 with neoantigens, 171 without neoantigens) are grouped into those with immunogenic neoantigens and those with none. Hazard ratios and log-rank *p*-values compare immunogenic patient tumors to non-immunogenic patient tumors. (**B**) The overall survival differences of stage I patients were classified based on the presence of immunogenic neoantigens and the number of altered genes. Immunogenic and high-NAG patient tumors (green, 37 patients) vs. non-immunogenic and low-NAG patient tumors (red, 23 patients) show HR = 0.0807, *p* = 7.8 × 10^−5^. Immunogenic and high-NAG tumors (green) vs. non-immunogenic and high-NAG patient tumors (blue, 147 patients) show HR = 0.229, *p* = 0.013. Non-immunogenic and high-NAG tumors (blue) vs. non-immunogenic and low-NAG patient tumors (red) show HR = 0.309, *p* = 7.9 × 10^−5^. High NAG patients with and without neoantigens were contrasted based on the relative abundance of immune cell types estimated by the TIMER algorithm (**C**), and the RNA expression of cytotoxic T-cell markers (**D**). Significant differences for each component are marked (*t*-test *p*-value * <0.05; ** <0.01).

**Figure 4 cancers-11-01009-f004:**
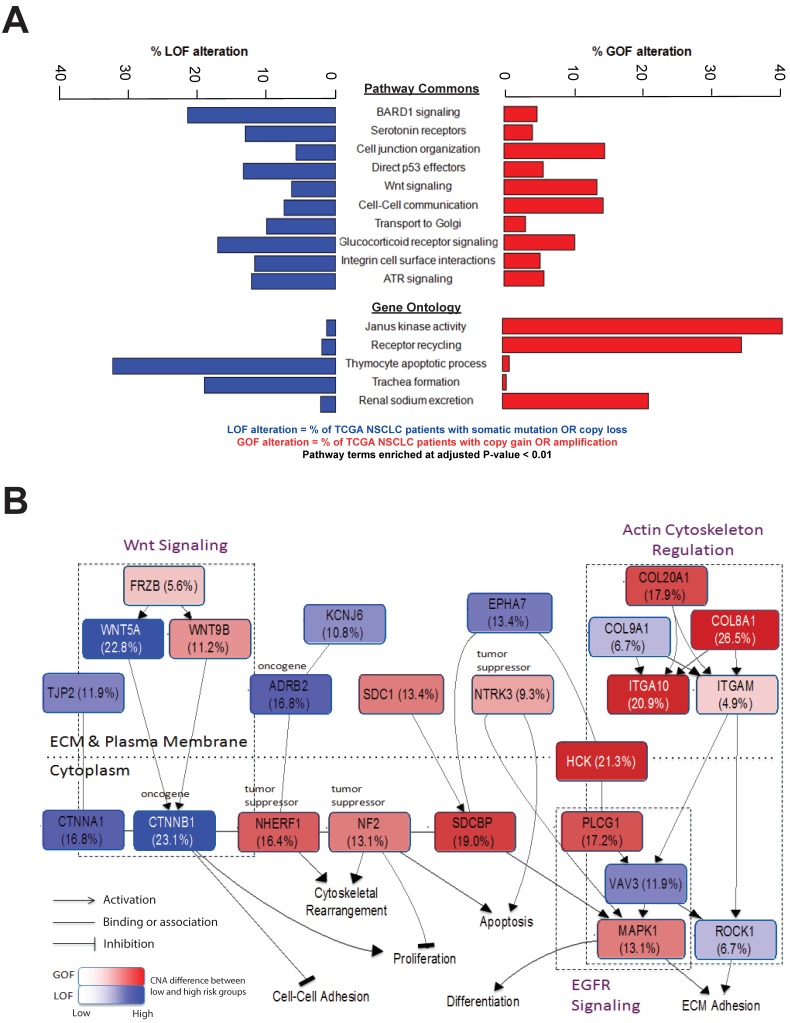
Enriched pathways and biological processes for genes in NAG. (**A**) Pathways in Pathway Commons and Gene Ontology were enriched for NAG genes. Bar plots show the proportions of NAG genes within each pathway that were classified as gain of function (GOF) or loss of function (LOF) alterations in TCGA cases. (**B**) A diagram of enriched pathways identified from NAG and their associated expression signature genes. The main pathways that emerged included Wnt signaling, integrin signaling and actin cytoskeleton regulation. Blue nodes identify copy number loss or inactivating mutation indicating loss of function (LOF), and red nodes indicate copy number gain indicating gain of function (GOF). Percentages in brackets represent the difference in alteration frequency between patients of low and high mortality risk, respectively, as classified by the NAG and the NAG expression score for TCGA NSCLC tumors. The color intensity reflects the proportion of tumors with the gene alteration. ECM = extracellular matrix.

**Figure 5 cancers-11-01009-f005:**
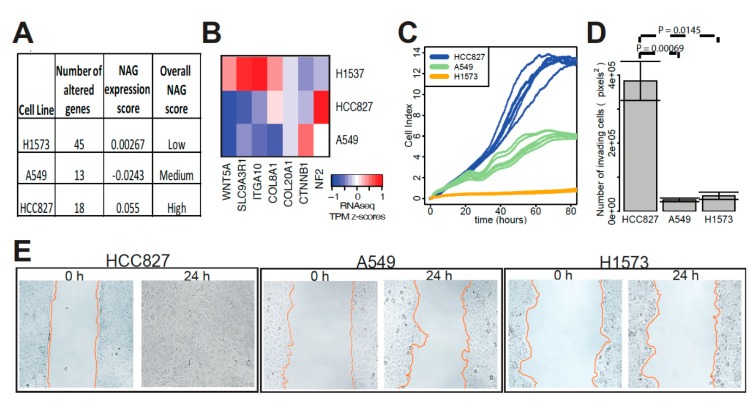
Functional impact of altered and differentially-expressed genes on cancer cells. (**A**) The number of genes (out of 865 genes) with somatic SNV or CNA was tallied for routinely-used lung cancer cell lines. Three lung cancer cell lines with differing overall NAG (based on quartiles) and different risk scores for NAG associated expression signature were compared to each other in functional assays. (**B**) A heatmap of RNA expression level scaled across cell lines for NAG genes commonly altered in TCGA is shown for the three lung cancer cell lines. (**C**) Proliferation of the three cultured cell lines was estimated by electrical impedance (cell index) over time. (**D**) Invasion of cells across trans-well membranes coated with Matrigel was counted. (**E**) Cell motility was measured by a time-lapse wound-healing assay. The rates of wound area travelled by cells were 15.42 µm/hr for HCC827, 2.64 µm/h for A549 and 1.98 µm/h for H1573.

**Table 1 cancers-11-01009-t001:** Multivariate survival model of NAG score with clinical-pathological factors of TCGA NSCLC stage I patients.

Variation	HR	95% CI	Wald Test *p*-Value
Age (>65 vs. ≤65)	1.04	0.62–1.73	0.89
Sex (F vs. M)	0.85	0.51–1.43	0.54
Tobacco (smoker vs. never)	1.75	0.74–4.12	0.2
Histology (adeno vs. squamous)	0.96	0.52–1.79	0.91
Overall NAG score (low vs. high burden)	2.46	1.27–4.75	0.0077 *

NAG, number of altered genes; HR, hazard ratio; CI, confidence interval; * designates significance at *p* ≤ 0.05.

**Table 2 cancers-11-01009-t002:** Multivariate survival model of the immunogenicity factor with clinical-pathological factors of TCGA NSCLC stage I patients.

Variation	HR	95% CI	Wald Test *p*-Value
Age of diagnosis (≤65 y vs. >65 y)	0.929	0.545–1.583	0.7865
Gender (male vs. female)	0.871	0.524–1.447	0.5929
Smoking history (last 15 y; yes vs. no)	0.751	0.434–1.302	0.3085
Histology (adeno vs. squamous cell)	0.677	0.392–1.169	0.1615
Immunogenicity (≥1 neoantigen vs. 0 neoantigens)	0.296	0.119–0.740	0.00919 *

* designates significance at *p* ≤ 0.05.

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
