# Peer review of "Somatic Alteration Burden Involving Non-Cancer Genes Predicts Prognosis in Early-Stage Non-Small Cell Lung Cancer"

_cancers, 2019, doi:10.3390/cancers11071009_

Round 1
Reviewer 1 Report
The authors identified genomic features of NSCLC tumor cells that could confer prognostic implication in patient clinical outcome. The results are very interesting and might be helpful to predict the prognosis of NSCLC patients. The methodology to demonstrate their hypothesis is appropriate, and diverse approach to verify results is impressive. However, there are several concerns that the authors need to address in order to publish an interpretable story for the reader.
1. In line 184, HC827 should be corrected into HCC827.
2. Please add some explanation how the authors estimated the overall NAG score in Figure 5A.
3. Please reveal the medium condition used in wound-healing assay (Figure 5D). If FBS was added in the medium, it could affect to the results in Figure 5D because three cell lines showed different proliferation ability in Figure 5B.
4. Please add results in Figure 5 about the different expression level of several genes mentioned in line 176-178 among the three cell lines.
Author Response
Reviewer 1
The authors identified genomic features of NSCLC tumor cells that could confer prognostic implication in patient clinical outcome. The results are very interesting and might be helpful to predict the prognosis of NSCLC patients. The methodology to demonstrate their hypothesis is appropriate, and diverse approach to verify results is impressive. However, there are several concerns that the authors need to address in order to publish an interpretable story for the reader.
We are glad you enjoyed reading our manuscript and found it interesting. Your comments on improving the interpretability of the story was very helpful and we implemented all the edits you suggested.
1. In line 184, HC827 should be corrected into HCC827.
HC827 corrected into HCC827
2. Please add some explanation how the authors estimated the overall NAG score in Figure 5A.
We added the following explanation to the Figure 5A caption to explain the overall NAG score in cell lines: “The number of genes (out of 865 genes) with somatic SNV or CNA was tallied for routinely used lung cancer cell lines. Three lung cancer cell lines with differing overall NAG (based on quartiles) and different risk scores for NAG associated expression signature were compared to each other in functional assays.”
3. Please reveal the medium condition used in wound-healing assay (Figure 5D). If FBS was added in the medium, it could affect to the results in Figure 5D because three cell lines showed different proliferation ability in Figure 5B.
We appreciate the reviewer's concern. The FBS was added to the medium and the migration assay was carried out for 24 hours. As Figure 5B is showing, there is no difference in proliferation between HCC828 and A549 cell lines at 24 hours. Therefore, the result of migration in Figure 5D is showing the difference in migration ability of these cells after 24 hours. This clarification was added to the Methods section.
4. Please add results in Figure 5 about the different expression level of several genes mentioned in line 176-178 among the three cell lines.
This is a great suggestion. We have now added the expression profiles in cell lines of several genes mentioned to be frequently altered in TCGA cases. All the genes except for COL20A1 seemed to have different levels of expression across the cell lines. This new result can be seen in a new Figure 5B and mentioned in the Results section.
Reviewer 2 Report
This is a very interesting and brilliantly written paper documenting the feasibility to assess non-small cell lung cancer prognosis at early stages, which may be critical for the decision-making process on individually-planned therapy. Authors appropriately use both dry and wet lab approach to confirm their hypotheses. Despite the high value of these novel findings, there seems to be a couple of formatting inconsistencies and other minor formal issues found within the text. Most importantly, several instances of nonstandard abbreviations/acronyms are not defined, which could make the manuscript less comprehensible for a general reader.
1) The semantic structure of the sentence "Simply counting the number of altered genes (NAG) among these 865 genes was 24 associated with longer disease-free survival (HR=0.153, p=1.48x10-4)." (line 24) may suggest that those who count NAG have longer disease-free survival. Hence, the authors may wish to rephrase this sentence to yield the desired meaning that NAG counts were associated with longer disease-free survival.
2) Please define abbreviations for PDX (line 60), CNA (line 61), SNV (line 61), TCGA (line 66), CNV (line 228), RSEM (line 274), ECM (line 343), FFPE (Supplementary Table 7), TMB (Supplementary Table 7), GLCI (Supplementary Table 7), and COSMIC (Supplementary Table 7) at their first occurrence.
3) The meaning of the sentence "3236 genes mutated in least one sample"; the codes "X169", "X178", "X154"...; the descriptors "X = recurred"; "mutant diploid", "wildtype, loss", "mutant, loss", "wildtype, gain", "mutant, gain" are not clear in Figure 2 (line 90). Could the general meaning be succinctly mentioned in the figure legend?
4) Similarly, could the authors add the meaning of the intervals (0.382 – 0.870) and (0.222 – 0.685) to the legend of Figure 2 (line 90)?
5) Sentence "Hierarchical clustering ordering with genes across the samples." (line 95) does not make sense as there is no verb in it.
6) It is stated that there were "moderately higher proportions of B-cells, dendritic cells, and CD8 T-cells in high NAG patient tumors with immunogenic peptides compared to without" (line 143), despite these differences are not highlighted as significant (*) for dendritic and CD8 T-cells in Figure 3C.
7) It is stated that there was "significantly higher expression of cytokines CCL5, CXCL9, CXCL10, and IL16", however this difference is not highlighted as significant (*) for IL16 in Figure 3D.
8) Figures 4A (line 188) and 4B (line 189) are difficult to read due to decreased image resolution.
9) Could the authors please incorporate a brief mention that PathwayCommons and GeneOntology databases were used to the legend of Figure 4A? Also, would it be possible to have equal axis scales for % of LOF and GOF alterations so that these values are directly comparable with each other in Figure 4A?
10) Would it be possible to supply Figure 4B with the definitions "ECM" and "CNV" acronyms?
11) The sentence "thus a marker for less aggressive tumors in patients with good survival outcomes" (line 242) does not make sense as it contains no verb.
12) Legends to Supplementary Tables (lines 366–377) could be sorted in increasing order (Supplementary Tables 5 and 6 after Supplementary Table 4, before 7).
13) What does "Illumina", "Affy133+2", "Affy 133A" in Supplementary Figures 1b-d refer to? Also, would it please be possible to have the panel labels to Supplementary Figure 1a-d and its figure legend corrected to uppercase (A-D) for increased consistency?
14) The color marking patients with immunogenic neoantigens in Supplementary Figure 2A looks more like teal than blue.
15) Please remove empty Sheets2 and 3 from Supplementary Table 4.
There are also few corrections suggested below. Arrow => points to the correct version and the corresponding adjustment is indicated within a pair of angle brackets <>.
1) line 2: Somatic alterations burden => Somatic alteration burden <singular>
2) line 15: Correspondence:Address => Correspondence: Address <space>
3) line 25: p=1.48x10-4 => P=1.48x10-4 <uppercase P>
4) line 26: were => was <singular>
5) line 32: burden;copy => burden; copy <space>
6) line 39: genes and gene => genes, and gene <comma>
7) line 55: patients, and validated => patients and validated <no comma>
8) line 65: GSE42127 and GSE68465 => GSE42127, and GSE68465 <comma>
9) line 75: HR= 0.366 => HR=0.366 <no space>
10) line 75: CI=0.140-0.952 => CI=0.140 – 0.952 <space 2x, dash>
11) line 76: CI=0.103-0.840 => CI=0.103 – 0.840 <space 2x, dash>
12) line 77: CI=0.185-1.61 => CI=0.185 – 1.61 <space 2x, dash>
13) line 78: CI=0.197-1.43 => CI=0.197 – 1.43 <space 2x, dash>
14) line 79: CI=0.179-1.79 => CI=0.179 – 1.79 <space 2x, dash>
15) line 80: CI=0.407-4.76 => CI=0.407 – 4.76 <space 2x, dash>
16) line 87: CI=0.051-0.459 => CI=0.051 – 0.459 <space 2x, dash>
17) line 90 (Figure 2C): HR = 0.391 (0.222 – 0.685 ) => HR = 0.391 (0.222 – 0.685) <no space>
18) line 96: in (B) 96 all TCGA NSCLC patients and (C) only in stage I patients. . => in 96 all TCGA NSCLC patients (B) and only in stage I patients (C). <patients (B), patients (C), only one period>
19) line 109: CI=0.332-0.952 => CI=0.332 – 0.952 <space 2x, dash>
20) line 110: CI=0.338-0.794 => CI=0.338 – 0.794 <space 2x, dash>
21) line 116: cut-offs that was used => cut-offs that were used <plural>
22) line 118: CI= 0.382 - 0.870 => CI=0.382 – 0.870 <no space, dash>
23) line 119: CI= 0.222 - 0.685 => CI=0.222 – 0.685 <no space, dash>
24) line 121: CI= 0.211 - 0.787 => CI=0.211 – 0.787 <no space, dash>
25) line 123: Stage => stage <lowercase s>
26) line 123: P-value => p-value <lowercase P>
27) line 123: >65 vs. <=65 => >65 vs <=65 <no period>
28) line 123: 0.62-1.73 => 0.62 – 1.73 <space 2x, dash>
29) line 123: F vs. M) => F vs M) <no period>
30) line 123: 0.51-1.43 => 0.51 – 1.43 <space 2x, dash>
31) line 123: 0.74-4.12 => 0.74 – 4.12 <space 2x, dash>
32) line 123: Adeno vs. Squamous => adeno vs squamous <lowercase a, s, no period>
33) line 123: 0.52-1.79 => 0.52 – 1.79 <space 2x, dash>
34) line 123: 1.27-4.75 => 1.27 – 4.75 <space 2x, dash>
35) line 124: signatures. => signatures <no period>
36) line 131: alleles, and => alleles and <no comma>
37) line 132: CI=0.341 - 0.8419 => CI=0.341 – 0.8419 <dash>
38) line 134: CI=0.1068 - 0.6619 => CI=0.1068 – 0.6619 <dash>
39) line 140: CI=0.02315 - 0.2813 => CI=0.02315 – 0.2813 <dash>
40) line 145: compared to without => compared to those without neoantigens <those, neoantigens>
41) line 151: Supplementary Figure S2B-C => Supplementary Figure S2B–C <dash>
42) line 153: patients. . => patients. <only one period>
43) line 153: 0.545 - 1.583 => 0.545 – 1.583 <dash>
44) line 153: 0.524 - 1.447 => 0.524 – 1.447 <dash>
45) line 153: 0.434 - 1.302 => 0.434 – 1.302 <dash>
46) line 153: 0.392 - 1.169 => 0.392 – 1.169 <dash>
47) line 153: 0.119 - 0.740 => 0.119 – 0.740 <dash>
48) line 157: neoantigens), are grouped => neoantigens) are grouped <no comma>
49) line 162: HR= 0.0807 => HR=0.0807 <no space>
50) line 163: HR= 0.229 => HR=0.229 <no space>
51) line 165: HR= 0.309 => HR=0.309 <no space>
52) line 165: were contrasted based on (C) the relative abundance of immune cell types estimated by the TIMER algorithm, and (D) the RNA expression of cytotoxic T-cell markers. => were contrasted based on the relative abundance of immune cell types estimated by the TIMER algorithm (C), and the RNA expression of cytotoxic T-cell markers (D). <algorithm (C), markers (D)>
53) line 173: extra-cellular => extracellular <no hyphen>
54) line 180: functions, such as => functions such as <no comma>
55) line 181: these pathways alterations => these pathway alterations <singular>
56) line 183: had 13 and 18 respectively => had 13 and 18, respectively <comma>
57) lines 184, 207: HC827 => HCC827
58) line 186: H827 => HCC827
59) line 193: Wnt Signaling, Integrin signaling and Actin Cytoskeleton Regulation => Wnt signaling, integrin signaling and actin cytoskeleton regulation <lowercase s, i, a, c, r>
60) line 195: (LOF), and => (LOF) and <no comma>
61) line 206: 15.42µm/hr for 206 HC827, 2.64µm/hr for A549, and 1.98µm/hr => 15.42 µm/hr for 206 HC827, 2.64 µm/hr for A549, and 1.98 µm/hr <space 3x>
62) line 211: nonsynonymous => non-synonymous <hyphen>
63) line 215: NAG and => NAG, and <comma>
64) line 234: remain => remains <singular>
65) line 235: impedes => impede <plural>
66) line 237: motility and => motility, and <comma>
67) line 248: Previously mutation => Previously, mutation <comma>
68) line 252: as high alteration burden and having long OS => as having high alteration burden and long OS <as having>
69) line 259: studies, and => studies and <no comma>
70) line 271: LUAD[42] => LUAD [42] <space>
71) line 274: RNAseq => RNA-seq <hyphen>
72) line 286: high vs. low => high vs low <no period>
73) line 291: number of altered gene => number of altered genes <plural>
74) line 301: cross validated => cross-validated <hyphen>
75) line 310: [1], and => [1] and <no comma>
76) line 313: Nonsynonymous => Non-synonymous <hyphen>
77) line 315: 8-11 => 8–11 <dash>
78) line 316, 317: 8-11mer => 8–11mer <dash>
79) line 320: Optitype[49] => Optitype [49] <space>
80) line 329: clonal and => clonal, and <comma>
81) line 337: ‘adhesion’, ‘integrin’ and ‘extracellular matrix’ => "adhesion", "integrin" and "extracellular matrix" <quotation mark 6x>
82) line 349: 8-µm => 8 µm <no hyphen>
83) line 358: et al have => et al. have <period>
84) line 364: NSCLC, through => NSCLC through <no comma>
85) line 372: genes. (see excel file Supplementary Table 1). => genes (see Excel file Supplementary Table 1). <no period, uppercase E>
86) lines 373, 374, 376: exel => Excel <uppercase E>
87) lines 378, 379: N-A.P. => N.-A.P. <period>
88) line 378: M.S.T => M.-S.T. <hyphen, period>
89) line 379: are involved => were involved <past tense>
90) line 379: analysis and interpretation => analysis, and interpretation <comma>
91) line 380: M.S.T. => M.-S.T. <hyphen>
92) line 383: #701595 and Canadian => #701595, and Canadian <comma>
93) line 387: Jessica Weiss and Ni Liu => Jessica Weiss, and Ni Liu <comma>
94) Supplementary Figure 1: Kaplan Meier curves showing => Kaplan-Meier curves show <hyphen, show>
95) Supplementary Figure 2: Stage I patients => stage I patients <lowercase s>
96) Supplementary Figure 2: (B) the relative abundance of immune cell types estimated by the TIMER algorithm and (C) the RNA expression of cytotoxic T-cell markers. => the relative abundance of immune cell types estimated by the TIMER algorithm (B) and the RNA expression of cytotoxic T-cell markers (C). <algorithm (B), no comma, markers (C)>
97) Supplementary Figure 3: panel only and immunogenic missense mutations => panel only, and immunogenic missense mutations <comma>
98) Supplementary Table 2 => Supplementary Table 2: <colon and bold formatting>
99) Supplementary Table 3 => Supplementary Table 3: <colon>
100) List of 36 NSCLC cases in the discovery set characterized by NAG levels and assoicated clinic-pathological features => List of 36 NSCLC cases in the discovery set characterized by NAG levels and assoicated clinic-pathological features <no bold formatting>
101) Supplementary Table 5: Supplementary Table 5 => Supplementary Table 5. <period>
102) Supplementary Table 6: Supplementary Table 6: => Supplementary Table 5. <period>
Author Response
Reviewer 2
This is a very interesting and brilliantly written paper documenting the feasibility to assess non-small cell lung cancer prognosis at early stages, which may be critical for the decision-making process on individually-planned therapy. Authors appropriately use both dry and wet lab approach to confirm their hypotheses. Despite the high value of these novel findings, there seems to be a couple of formatting inconsistencies and other minor formal issues found within the text. Most importantly, several instances of nonstandard abbreviations/acronyms are not defined, which could make the manuscript less comprehensible for a general reader.
We very much appreciate your comments and suggestions. Everything you suggested was sensible and we have made all the edits requested.
1) The semantic structure of the sentence "Simply counting the number of altered genes (NAG) among these 865 genes was 24 associated with longer disease-free survival (HR=0.153, p=1.48x10-4)." (line 24) may suggest that those who count NAG have longer disease-free survival. Hence, the authors may wish to rephrase this sentence to yield the desired meaning that NAG counts were associated with longer disease-free survival.
We have rephrased this statement to “Simply, the number of altered genes (NAG) among these 865 genes was associated with longer disease-free survival (HR=0.153, P=1.48x10-4).”
2) Please define abbreviations for PDX (line 60), CNA (line 61), SNV (line 61), TCGA (line 66), CNV (line 228), RSEM (line 274), ECM (line 343), FFPE (Supplementary Table 7), TMB (Supplementary Table 7), GLCI (Supplementary Table 7), and COSMIC (Supplementary Table 7) at their first occurrence.
We have made the following corrections to defining the abbreviations:
Figure 1 caption title: PDXs -> patient-derived xenografts (PDXs)
USA spelling used throughout for consistency
I think it would be worth defining PDXs along with showing the acronym in the introduction (at the moment these are mentioned in the introduction but the term patient-derived xenograft/PDX isn’t used in the text of the introduction) and again in the methods since this term gets used a lot and might not be widely known.
Figure 1 caption (A): CNA -> copy number aberration (CNA)
Figure 1 caption (A): point mutations (SNV) -> single nucleotide variant (SNV)
Figure 1 caption: (TCGA) -> from The Cancer Genome Atlas (TCGA),
Line 228: CNV -> CNA
SNV used in place of point mutation throughout the text
Line 274: RSEM scale -> RNA-Seq by Expectation Maximization (RSEM) scale
Line 343: ECM -> extracellular matrix (ECM)
Figure 4B caption: ECM defined
Supplementary Table 7: FFPE -> formalin-fixed, paraffin-embedded (FFPE)
Supplementary Table 7: TMB -> tumor mutation burden (TMB)
Supplementary Table 7: GLCI -> Great Lakes Cancer Institute (GLCI)
Supplementary Table 7: COSMIC -> the Catalogue Of Somatic Mutations In Cancer (COSMIC)
3) The meaning of the sentence "3236 genes mutated in least one sample"; the codes "X169", "X178", "X154"...; the descriptors "X = recurred"; "mutant diploid", "wildtype, loss", "mutant, loss", "wildtype, gain", "mutant, gain" are not clear in Figure 2 (line 90). Could the general meaning be succinctly mentioned in the figure legend?
Thank you for pointing out the confusing labeling. In the Figure 2 caption, we have now clarified the 3236 genes, sample codes designating PDXs (eg. X169), and the different combinations of point mutations and copy alterations (eg. mutant, loss).
4) Similarly, could the authors add the meaning of the intervals (0.382 – 0.870) and (0.222 – 0.685) to the legend of Figure 2 (line 90)?
We have now clarified in the Figure 2 caption that these intervals are the confidence intervals for the hazard ratios.
5) Sentence "Hierarchical clustering ordering with genes across the samples." (line 95) does not make sense as there is no verb in it.
We have rephrased this to “Genes were ordered by hierarchical clustering across the samples”.
6) It is stated that there were "moderately higher proportions of B-cells, dendritic cells, and CD8 T-cells in high NAG patient tumors with immunogenic peptides compared to without" (line 143), despite these differences are not highlighted as significant (*) for dendritic and CD8 T-cells in Figure 3C.
Thank you for pointing this out and you are correct that these differences were not labeled as significant in Figure 3C. That is why we stated “moderately higher” as referring to the effect size instead of the statistical singificance. We felt it was still important to report the difference despite the high p-value, because the uncertainty in immune cell estimates from RNAseq is large and we wanted to encourage larger studies that are powered to validate or refute our findings. To avoid confusion and not to be misleading, we have now added to the result statement that the difference was “not significant”.
7) It is stated that there was "significantly higher expression of cytokines CCL5, CXCL9, CXCL10, and IL16", however this difference is not highlighted as significant (*) for IL16 in Figure 3D.
We were citing a paper that found association between the cytokines and cytotoxic T-cells, which aligned to our observation of higher cytokine expression in our immunogenic group. We observed that all the cytokines except IL16 had significantly higher expression. We have now separated the citation of the previous study with our observation to make it clear that IL16 was not validated by our study.
8) Figures 4A (line 188) and 4B (line 189) are difficult to read due to decreased image resolution.
Apologies, we have now uploaded a high resolution image into the manuscript document.
9) Could the authors please incorporate a brief mention that PathwayCommons and GeneOntology databases were used to the legend of Figure 4A? Also, would it be possible to have equal axis scales for % of LOF and GOF alterations so that these values are directly comparable with each other in Figure 4A?
Brief mention and explanation of how PathwayCommons and GeneOntology was used in Figure 4A has now been added to the figure caption. We have made equal axis scales for % LOF and GOF in Figure 4A.
10) Would it be possible to supply Figure 4B with the definitions "ECM" and "CNV" acronyms?
“CNV” was changed to “CNA” to be consistent with term used in other parts of the manuscript. The definition of ECM was put in the caption.
11) The sentence "thus a marker for less aggressive tumors in patients with good survival outcomes" (line 242) does not make sense as it contains no verb.
“And is” inserted in front of “thus” sentence to correct this.
12) Legends to Supplementary Tables (lines 366–377) could be sorted in increasing order (Supplementary Tables 5 and 6 after Supplementary Table 4, before 7).
Supplementary Table legends resorted in numerical order.
13) What does "Illumina", "Affy133+2", "Affy 133A" in Supplementary Figures 1b-d refer to? Also, would it please be possible to have the panel labels to Supplementary Figure 1a-d and its figure legend corrected to uppercase (A-D) for increased consistency?
We have clarified in the caption of Supplementary Figure 1 that “Illumina” refers to RNAseq, "Affy133+2" and "Affy 133A" refer to microarray platforms used to measure expression in those cohorts. We have also edited Supplementary Figure 1 to make panel labels all uppercase for consistency.
14) The color marking patients with immunogenic neoantigens in Supplementary Figure 2A looks more like teal than blue.
Caption changed “blue” -> “teal” to reflect this.
15) Please remove empty Sheets2 and 3 from Supplementary Table 4.
Sheets2 and 3 from Supplementary Table 4 have been removed.
There are also few corrections suggested below. Arrow => points to the correct version and the corresponding adjustment is indicated within a pair of angle brackets <>.
Thank you for providing these helpful suggestions. All suggested corrections below have been made to the manuscript and supplementary materials.
1) line 2: Somatic alterations burden => Somatic alteration burden <singular>
2) line 15: Correspondence:Address => Correspondence: Address <space>
3) line 25: p=1.48x10-4 => P=1.48x10-4 <uppercase P>
4) line 26: were => was <singular>
5) line 32: burden;copy => burden; copy <space>
6) line 39: genes and gene => genes, and gene <comma>
7) line 55: patients, and validated => patients and validated <no comma>
8) line 65: GSE42127 and GSE68465 => GSE42127, and GSE68465 <comma>
9) line 75: HR= 0.366 => HR=0.366 <no space>
10) line 75: CI=0.140-0.952 => CI=0.140 – 0.952 <space 2x, dash>
11) line 76: CI=0.103-0.840 => CI=0.103 – 0.840 <space 2x, dash>
12) line 77: CI=0.185-1.61 => CI=0.185 – 1.61 <space 2x, dash>
13) line 78: CI=0.197-1.43 => CI=0.197 – 1.43 <space 2x, dash>
14) line 79: CI=0.179-1.79 => CI=0.179 – 1.79 <space 2x, dash>
15) line 80: CI=0.407-4.76 => CI=0.407 – 4.76 <space 2x, dash>
16) line 87: CI=0.051-0.459 => CI=0.051 – 0.459 <space 2x, dash>
17) line 90 (Figure 2C): HR = 0.391 (0.222 – 0.685 ) => HR = 0.391 (0.222 – 0.685) <no space>
18) line 96: in (B) 96 all TCGA NSCLC patients and (C) only in stage I patients. . => in 96 all TCGA NSCLC patients (B) and only in stage I patients (C). <patients (B), patients (C), only one period>
19) line 109: CI=0.332-0.952 => CI=0.332 – 0.952 <space 2x, dash>
20) line 110: CI=0.338-0.794 => CI=0.338 – 0.794 <space 2x, dash>
21) line 116: cut-offs that was used => cut-offs that were used <plural>
22) line 118: CI= 0.382 - 0.870 => CI=0.382 – 0.870 <no space, dash>
23) line 119: CI= 0.222 - 0.685 => CI=0.222 – 0.685 <no space, dash>
24) line 121: CI= 0.211 - 0.787 => CI=0.211 – 0.787 <no space, dash>
25) line 123: Stage => stage <lowercase s>
26) line 123: P-value => p-value <lowercase P>
27) line 123: >65 vs. <=65 => >65 vs <=65 <no period>
28) line 123: 0.62-1.73 => 0.62 – 1.73 <space 2x, dash>
29) line 123: F vs. M) => F vs M) <no period>
30) line 123: 0.51-1.43 => 0.51 – 1.43 <space 2x, dash>
31) line 123: 0.74-4.12 => 0.74 – 4.12 <space 2x, dash>
32) line 123: Adeno vs. Squamous => adeno vs squamous <lowercase a, s, no period>
33) line 123: 0.52-1.79 => 0.52 – 1.79 <space 2x, dash>
34) line 123: 1.27-4.75 => 1.27 – 4.75 <space 2x, dash>
35) line 124: signatures. => signatures <no period>
36) line 131: alleles, and => alleles and <no comma>
37) line 132: CI=0.341 - 0.8419 => CI=0.341 – 0.8419 <dash>
38) line 134: CI=0.1068 - 0.6619 => CI=0.1068 – 0.6619 <dash>
39) line 140: CI=0.02315 - 0.2813 => CI=0.02315 – 0.2813 <dash>
40) line 145: compared to without => compared to those without neoantigens <those, neoantigens>
41) line 151: Supplementary Figure S2B-C => Supplementary Figure S2B–C <dash>
42) line 153: patients. . => patients. <only one period>
43) line 153: 0.545 - 1.583 => 0.545 – 1.583 <dash>
44) line 153: 0.524 - 1.447 => 0.524 – 1.447 <dash>
45) line 153: 0.434 - 1.302 => 0.434 – 1.302 <dash>
46) line 153: 0.392 - 1.169 => 0.392 – 1.169 <dash>
47) line 153: 0.119 - 0.740 => 0.119 – 0.740 <dash>
48) line 157: neoantigens), are grouped => neoantigens) are grouped <no comma>
49) line 162: HR= 0.0807 => HR=0.0807 <no space>
50) line 163: HR= 0.229 => HR=0.229 <no space>
51) line 165: HR= 0.309 => HR=0.309 <no space>
52) line 165: were contrasted based on (C) the relative abundance of immune cell types estimated by the TIMER algorithm, and (D) the RNA expression of cytotoxic T-cell markers. => were contrasted based on the relative abundance of immune cell types estimated by the TIMER algorithm (C), and the RNA expression of cytotoxic T-cell markers (D). <algorithm (C), markers (D)>
53) line 173: extra-cellular => extracellular <no hyphen>
54) line 180: functions, such as => functions such as <no comma>
55) line 181: these pathways alterations => these pathway alterations <singular>
56) line 183: had 13 and 18 respectively => had 13 and 18, respectively <comma>
57) lines 184, 207: HC827 => HCC827
58) line 186: H827 => HCC827
59) line 193: Wnt Signaling, Integrin signaling and Actin Cytoskeleton Regulation => Wnt signaling, integrin signaling and actin cytoskeleton regulation <lowercase s, i, a, c, r>
60) line 195: (LOF), and => (LOF) and <no comma>
61) line 206: 15.42µm/hr for 206 HC827, 2.64µm/hr for A549, and 1.98µm/hr => 15.42 µm/hr for 206 HC827, 2.64 µm/hr for A549, and 1.98 µm/hr <space 3x>
62) line 211: nonsynonymous => non-synonymous <hyphen>
63) line 215: NAG and => NAG, and <comma>
64) line 234: remain => remains <singular>
65) line 235: impedes => impede <plural>
66) line 237: motility and => motility, and <comma>
67) line 248: Previously mutation => Previously, mutation <comma>
68) line 252: as high alteration burden and having long OS => as having high alteration burden and long OS <as having>
69) line 259: studies, and => studies and <no comma>
70) line 271: LUAD[42] => LUAD [42] <space>
71) line 274: RNAseq => RNA-seq <hyphen>
72) line 286: high vs. low => high vs low <no period>
73) line 291: number of altered gene => number of altered genes <plural>
74) line 301: cross validated => cross-validated <hyphen>
75) line 310: [1], and => [1] and <no comma>
76) line 313: Nonsynonymous => Non-synonymous <hyphen>
77) line 315: 8-11 => 8–11 <dash>
78) line 316, 317: 8-11mer => 8–11mer <dash>
79) line 320: Optitype[49] => Optitype [49] <space>
80) line 329: clonal and => clonal, and <comma>
81) line 337: ‘adhesion’, ‘integrin’ and ‘extracellular matrix’ => "adhesion", "integrin" and "extracellular matrix" <quotation mark 6x>
82) line 349: 8-µm => 8 µm <no hyphen>
83) line 358: et al have => et al. have <period>
84) line 364: NSCLC, through => NSCLC through <no comma>
85) line 372: genes. (see excel file Supplementary Table 1). => genes (see Excel file Supplementary Table 1). <no period, uppercase E>
86) lines 373, 374, 376: exel => Excel <uppercase E>
87) lines 378, 379: N-A.P. => N.-A.P. <period>
88) line 378: M.S.T => M.-S.T. <hyphen, period>
89) line 379: are involved => were involved <past tense>
90) line 379: analysis and interpretation => analysis, and interpretation <comma>
91) line 380: M.S.T. => M.-S.T. <hyphen>
92) line 383: #701595 and Canadian => #701595, and Canadian <comma>
93) line 387: Jessica Weiss and Ni Liu => Jessica Weiss, and Ni Liu <comma>
94) Supplementary Figure 1: Kaplan Meier curves showing => Kaplan-Meier curves show <hyphen, show>
95) Supplementary Figure 2: Stage I patients => stage I patients <lowercase s>
96) Supplementary Figure 2: (B) the relative abundance of immune cell types estimated by the TIMER algorithm and (C) the RNA expression of cytotoxic T-cell markers. => the relative abundance of immune cell types estimated by the TIMER algorithm (B) and the RNA expression of cytotoxic T-cell markers (C). <algorithm (B), no comma, markers (C)>
97) Supplementary Figure 3: panel only and immunogenic missense mutations => panel only, and immunogenic missense mutations <comma>
98) Supplementary Table 2 => Supplementary Table 2: <colon and bold formatting>
99) Supplementary Table 3 => Supplementary Table 3: <colon>
100) List of 36 NSCLC cases in the discovery set characterized by NAG levels and assoicated clinic-pathological features => List of 36 NSCLC cases in the discovery set characterized by NAG levels and assoicated clinic-pathological features <no bold formatting>
101) Supplementary Table 5: Supplementary Table 5 => Supplementary Table 5. <period>
102) Supplementary Table 6: Supplementary Table 6: => Supplementary Table 5. <period>
Round 2
Reviewer 1 Report
I think the manuscript is now appropriate to be published. Thank you.